# Experimental Investigation of the Tensile Behavior of Selected Tire Cords Using Novel Testing Equipment

**DOI:** 10.3390/ma15124163

**Published:** 2022-06-12

**Authors:** Paweł Bogusz, Danuta Miedzińska, Marcin Wieczorek

**Affiliations:** Faculty of Mechanical Engineering, Military University of Technology, Kaliskiego 2 St., 00-908 Warsaw, Poland; pawel.bogusz@wat.edu.pl (P.B.); marcin.wieczorek@wat.edu.pl (M.W.)

**Keywords:** polyamide cords, tension test, elongation at break, breaking force

## Abstract

Aramid and polyamide cords are used in a wide range of applications, particularly in the automotive industry (tire reinforcement) and textile industry for military and fireguard purposes. The problem of the reliable experimental study of tensile behavior of synthetic cords is considered in this paper. In the available standards for synthetic cord testing, particularly ASTM D 885-03, the tensile test must result with the cord damage in the middle of gauge length, and the cords should be fixed in the machine clamps. The trial test gave damage near the clamps. We propose a novel testing stage mounted in the testing machine clamps to achieve the uniform tensile stress distribution in the gauge length of the measured cords. The results of the deformations were measured in two ways: using testing machine head displacement and a videoextensometer. Stress curves of four distinguished cords were evaluated and compared. The second method allowed to acquire results differing from the manufacturers’ data from 0.7% to 21.5%, which allowed for the conclusion that the designed test stand allows for obtaining reliable results for stretched cords.

## 1. Introduction

Polyamide and aramid (aromatic polyamide) yarns are synthetic materials well known due to their high strength and heat resistance [1,2]. Aramids in particular are selected from polyamides because of the aromatic groups appearing in their chains. Those fibers find a wide applications in a textile industry (for example textiles, such as Kevlar poly(phenylene-1,4-diamide) and Nomex poli (phenyle-1,3-diamide) [3], are used in fireman protective clothing [4,5]) as well as in military applications due to their high ballistic resistance [6,7].

Polyamide yarns can be used to produce pure materials; however, often they are applied as a reinforcement in composite materials. Such application was described by Tonatto, Forte and Amico [8]. The authors studied the tensile and fatigue behavior of rubbers reinforced with polyester, polyamide and hybrid polyaramid/polyamide cords to assess their strength and phenomena in the microstructure. Rubber matrix composites with PET (polyethylene terephthalate) cords were also examined by Carbone et al. for their micromechanical behavior assessment using laser Raman microscopy [9].

Qian et al. [10] tested the effects of the fiber content and strain rate on aramid fabric reinforced polyamide composite. Glass/PA 6 (polyamide 6) composites made of hybrid yarns (PA 6 fibers were coated with DT2 antistatic preparation) were investigated by Klata, Velda and Krucińska [11] due to their crystalline structure and its influence on material mechanical properties.

The interesting comparison of tensile behavior of the following materials: CFRP—carbon fiber reinforced polymer composite, AFRP—aramid fiber reinforced polymer composite and GFRP—glass fiber reinforced polymer composite) polymer composites with pre-stressing steel was presented by Giannopoulos and Burgoyne [12]. Testing the composites the authors achieved linear stress–strain curves when the steel behavior was typically nonlinear. The composites achieved a significantly higher breaking stress (CFRP—31 GPa, AFRP—29 GPa and GFRP—24 GPa) in comparison to steel (18 GPa); however, the breaking stress of composites was smaller (from 1.7% for CFRP to 3.1% for GFRP) than for steel (more than 6%).

Hengstermann et al. [13] presented the comparison of carbon fiber and aramid fiber composites in accordance to the fiber length and preparation on their mechanical properties. Another interesting application of aramid yarns and cords is their implementation to shear thickening fluids for improving the ballistic performance [14] and rheological response [15]. The newest application of polyamide and aramid fibers is 3D printing. They are used for example as PCL scaffold reinforcement [16] and filaments [17] or multifilaments [18].

One of the most important applications of synthetic cords is in the automotive industry for: interior fabric for automobiles [19], conveyor belts, tire cords and hoses [20,21,22]. The methods of mechanical testing of cords and fibers were widely described in the scientific literature. Tensile tests of HMPE (High Modulus PolyEthylene) fibers for their fatigue failure assessment were conducted by Humeau et al. [23]. The authors designed the special equipment for dynamic testing (Figure 1a). 

Tensile fatigue behavior of polyamide 66 nanofiber cords was studied by Mooneghi et al. using electrospinning setup (Figure 1b) [24]. The stress–strain and modulus–strain curves achieved in tensile tests of poly(ethylene terephthalate) and polyamide industrial yarns were presented by Jing and Shanyuan [25] with no description of method of fixing the yarn in the testing machine. The PVA (poly(vinyl acetate)) braided cords mechanical behavior using gripping systems (Figure 1c) was tested by Freire et al. [26] and Poly (lactic acid) Yarns with Added Rosins—using laboratory spinning equipment COLLIN^®^ CMF 100—by Bolskis et al. [27]—Figure 1d.

Generally, the most troublesome part of the tensile testing of cords is a method of fixing the cords in the testing machine. In the methods presented above, the special equipment was designed to wrap the cord and to measure its strength. Other authors also proposed the use of resin to fix the ends of the cord [28]—Figure 1e. The method of cord strain or elongation at break measurement can be difficult.

In strength investigations, various solutions for fixing cord materials are used in order to solve this issue. In the simplest solution both ends of the cord can be held directly in the clamps of the testing machine. The base of the strain (initial gauge length) is then well defined, however it typically causes premature break of the cord, the ends of which are damaged by the additional load generated by the jaws and often also the serrated structure of the clamps inserts. 

In order to prevent this effect, additional friction surfaces are used in a specifically designed gripping mechanism, and thus the cords wrapped on them no longer break in the clamps area; however, then, it is more difficult to define the initial gauge length for strain calculations. The gripping parts of the cord, wound up on the friction parts of the grips can constitute a large part of its total length, and they also participate in the elongation process.

In the investigation of the cords presented in this paper, grips with additional friction surfaces were used. In order to avoid the influence of long gripping parts of the cord on the strain measurement, the digital image correlation method (DIC) was used. The optical methods of strain measurement are widely used in a strength analysis. The methods allow evaluation of the true stress–strain curve of the investigated material. The digital image correlation method was used to measure the tensile curve and strength properties of the tire rubber material by Baranowski et al. [29], where a high-speed camera was used as a videoextensometer. They recorded the position of two appropriate markers installed in the measuring part of the cord.

The overall conditions of the aramid yarns tensile testing, which were applied in the presented study were stated in ASTM D885-03 and ASTM: D7269/D7269M-17 standards [30,31].

The aim of the study was to test the selected tire cords on the innovative particularly designed testing stage and to assess the difference of the achieved results in comparison to manufacturer’s data to check if the proposed equipment allows to obtain the correct results. The stress–strain curves of four distinguish cords were evaluated and compared. The results of elongation were measured in two ways: using testing machine head displacement sensor and videoextensometer.

## 2. Materials

Four types of synthetic tire cords were tested in the form of cut sections—Figure 2. Cords with the following trade names were tested: Kordarna Plus PA6 (Kordárna Plus, a.s., Velká nad Veličkou, Czech Republic), PA6_Cordenka (Cordenka GmbH & Co. KG, Obernburg, Germany), PES 1440 (special version, Kordárna Plus, a.s., Velká nad Veličkou, Czech Republic), Kordarna Plus PES (Kordárna Plus, a.s., Velká nad Veličkou, Czech Republic). The data of the PES 1440 cord are restricted because of its military application and company secret and the material was produced on a special order.

Kordarna Plus PA6 is a cord fabric made of polyamide 6 yarn, which is mainly used for the production of truck and agricultural tires (in the skeleton of the tire). PA6_Cordenka is a polyamide cord used for tire manufacturing process, in particular as a reinforcement in “street racing” tires. Kordarna Plus PES is a cord fabric produced from polyester yarn and used for the production of passenger tires and light cargo tires (in the skeleton of the tire). The detailed data on the composition and structure of the above-mentioned cords are proprietary data of their manufacturers (Kordarna and Cordenca companies).

The names, symbols and material parameters declared by the manufacturer of the tested cords were presented in Table 1.

## 3. Methods

A specifically designed testing stage of the grips were used to fasten the cord, as shown in Figure 3. They were designed to conduct tensile tests of various types of plastic threads and can be installed in the clamps of the universal testing machine as shown in Figure 4. Their use allowed to avoid premature breakage of threads, installed directly in the clamps of the testing machine.

The structure of the grips (Figure 3) allows them to be used for various types of mounting in a strength machines clamps. They have flat inserts (1) to jaws with flat clamps and threaded mounts (2). When installing the cord samples, their end fragments are wound on the rollers (3) and fixed in the clamps of the grips (4). The minimum angle of thread wrap on the rollers is 330°. The grip rollers have a working part diameter of 28 mm (circumference of 88 mm). The clamps of the grips have a working length of 34.6 mm.

The grip is designed so that the loading force is aligned with the axis of the sample. When the cord is tested, the sample is clamped on the rollers (3) while being tensed and held in the clamps (4). In this way, friction between the rollers and the ends of the cord is produced, which prevents the thread from breaking directly in the grips. 

The measurements of the displacement were conducted in two ways: directly from the machine head sensors and using the videoextensometer Phantom V12 Vision Research (Vision Research, Inc., New Jersey, USA), which was placed in front of the testing machine, perpendicular to the plain of the cord grips. Strain measurement was performed using digital image correlation (DIC) method with the Tema 3D software (Image Systems AB, Linköping, Sweden). Two markers were attached to the cord on its measurement area. The distance of the markers from each other was about 200 mm. The methodology was similar for all investigated materials.

The DIC measurement method is based on the principles of mechanics of continuous media [32], where the changes in dimensions and location of short sections are considered by localization of section ending points before (*A, B*) and after (*A′, B′*) deformation. The points are described in a Cartesian coordination system (*x, y, z*) by the formula [33]:(1)$$A^{\prime }=\left(x_{1},\;y_{1},z_{1}\right)=\left[x+u\left(A\right),y+v\left(A\right),z+w\left(A\right)\right]$$
(2)$$B^{\prime }=(x_{1}+dx_{1},y_{1}+dy_{1},z_{1}+dz_{1})\;\begin{array}{cc} & =[x+u\left(A\right)+u\left(B\right)-u\left(A\right)+dx,y+dx,y+v\left(A\right)\\ & +v\left(B\right)-v\left(A\right)+dy,z+w\left(A\right)+w\left(Q\right)-w\left(P\right)+dz] \end{array}$$
where *u, v* and *w* are the components of the displacement vector along *x*, *y* and *z* axis, respectively.

The length of *AB* and *A′B′* segments can be calculated as [33]:(3)$$\left|AB\right|=dx^{2}+dy^{2}+dz^{2}$$
(4)$$\left|A\prime B\prime \right|=dx_{1}^{2}+dy_{1}^{2}+dz_{1}^{2}$$

On the basis of the Equations (1)–(4), the components of the strain state in two dimensional coordination system are described as [33]:(5)$$\varepsilon_{xx}≅\frac{\partial u}{\partial x}+\frac{1}{2}\left[{\left(\frac{\partial u}{\partial x}\right)}^{2}+{\left(\frac{\partial v}{\partial x}\right)}^{2}\right]$$
(6)$$\varepsilon_{yy}≅\frac{\partial v}{\partial y}+\frac{1}{2}\left[{\left(\frac{\partial u}{\partial y}\right)}^{2}+{\left(\frac{\partial v}{\partial y}\right)}^{2}\right]$$
(7)$$\varepsilon_{xy}≅\frac{1}{2}\left(\frac{\partial u}{\partial y}+\frac{\partial v}{\partial x}\right)+\frac{1}{2}\left[\frac{\partial u}{\partial x}\frac{\partial u}{\partial y}+\frac{\partial v}{\partial x}\frac{\partial v}{\partial y}\right]$$

## 4. Testing Procedure

The aim of the research was to verify the designed grips and the method of tensile testing of the cords. To achieve this aim, the experimental study was conducted to determine the force–strain curve and selected strength parameters (breaking force, elongation at break and initial modulus). The reason for the development of the new stand was the results of the cord initial tensile tests conducted directly in the machine clamps, according to the ASTM D-4018 standard. The performed tests led to the breaking of the cord in most cases near the upper clamp or the removal of the cord from the used protective spacer. A satisfactory repeatability of the results was also not obtained. Moreover, the achieved form of thread breaking was unacceptable considering the ASTM D-885 standard.

The cords tensile tests were conducted under static tensile load on the Instron 8802 (instton Company, Norwood, USA) universal testing machine (Figure 5). The guidelines included in the ASTM D885-3 standard [30] were used in the tests. The tests were conducted at an ambient temperature of 24 °C and an air humidity of about 50%.

The cords tensile tests were performed by controlling the displacement of the lower piston of the testing machine. The load speed was 300 ± 0.2 mm/min. The machine automatically recorded the loading force and piston position as a function of time, with a data sampling rate of 50 Hz. An additional Instron integrated precision force measuring head with a range of ±5 kN was used in the tests (Figure 5). An initial force was set for each cord type. Its values were calculated on the basis of the ASTM standard guidelines with the proportion of 5 mN/tex [30]. General parameters of the testing machine settings were presented in Table 2.

The mounting procedure was identical for all samples. The total arc of contact was 1050° (wounding almost three times), which corresponded to a cord length of 264 mm. A piece of cord wound on a roll and clamped in the grips was about 320 mm long (measured on one side). The total length of the portion of the sample wound and clamped in the clamps prior to testing was approximately 900 mm.

The gauge length of all samples was set at 250 ± 0.3 mm. It was measured between the points of contact of the cord sample with the rollers of cord tension grips. The determined gauge length corresponded to the absolute displacement of the machine cylinder equal to 124.1 mm. Ten samples of each type were tested.

To identify the place of cord breaking on the cord samples, after mounting them on the measuring stand, the points of the cord contact with the rollers and the places where the clamps of the thread grips were clamped were marked with felt-tip pens in green and red, respectively (Figure 6). Indicatively, half the sample length was marked in blue.

## 5. Results and Discussion

The example tensile test results were presented in Figure 7 as the photos of the broken cord III. All cord broke in the middle of the gauge length as indicated in the ASTM standard [30].

On the basis of the measured values the stress–strain characteristics were prepared using cN/dtex units [34] and presented in Figure 8 where the results for measurements with the use of machine head sensors were marked in orange and with videoextensometer—in black. The samples that broke near the grips in accordance to standard regulations [30] were neglected.

At first, the displacement of the machine head was considered to calculate the cord strain. The results of this procedure did not correspond to the values presented by the cord producers (the elongation at break value was about two-times higher). This phenomenon was likely caused by measuring so-called “undefined” deformation, which should be considered in the section formed by the wound on the roll and from the roll to the grip. Thus, it was decided to measure the strain of the cord using the videoextensometer Phantom V12 Vision Research.

The results were similar to the producers’ values as well the data presented by Tian et al. [34] and Aytac et al. [35]. In Figure 9, the example sequence of three photos of cord I shot during strain measurement is shown.

The achieved stress–strain curves can be divided into three characteristic sections—marked in Figure 8 only for DIC measurements [36]. Sector I can be assumed as the initial stage in which the tightening of the cord threads appears. Sector II is narrow, the twist effect of each cord is visible (the greater the twist was, the narrower the curve recorded, and the cord threads worked in no other effects). In sector III, the cords material structural properties influence the cord behavior [37].

The initial modulus of each cord was calculated on the basis of the force-displacement curves achieved using DIC method using the ASTM D 885-03 Standard guidelines (Figure 10). The actual cord linear density was also determined by the recommendations of ASTM D-885, using a linear ruler with a reading accuracy of 0.1% and a balance with an accuracy of 0.1 mg. With the obtained measurement results, the data of the devices used and the fact that it was an indirect uncorrelated measurement when estimating the total expanded uncertainty (coverage factor k = 2), the law of uncertainty propagation was used, obtaining a relative value equal to ± 1%. The results are presented in Conclusion section.

The DIC methods require the length measurement to be scaled to compute the displacement. In principle, scaling is not necessary to measure the deformation as it is a dimensionless quantity [pixel/pixel]. However, calibration of the lens distortion is required, particularly with 3D measurements.

During the DIC tests, the displacement was also determined. At that time, the displacement scaling by the DIC method was used for the movement of the head of the machine clamp with a measuring mark (checkerboard) glued on it. The known value of the head position was a measure of the assumed initial measuring length of the sample—250 mm. The DIC method was scaled in each test separately.

The difference in deformation measurements between the two methods seen in Figure 9 is not due to a scaling error but to triple winding of the thread ends on both sides on the rollers and in no case to the DIC measurement error. The DIC measurement error is in the tenths of a millimeter, hundredths of a pixel and is due to the quality of the markers and lighting. The measuring length of 250 mm, which was introduced when measuring the deformation from the machine head displacement, did not take into account the work of the ends of the cords wound on the rollers, what effected with the overestimated deformation and the noted difference.

It must be noticed that, for each cord testing, the statistical analysis of the results was conducted. The values of standard deviation and uncertainty with 95% of confidence were presented in Table 3. The uncertainty was calculated based on t-Student’s distribution with a 95% confidence, where the average value equals to [38,39]:(8)$$\bar{x}=\frac{1}{n}\overset{n}{\underset{i=1}{\sum }}x_{i}$$
where $x_{i}$ is a value of i-th measurement and n is a number of all measurements.

The standard deviation s was calculated as:(9)$$s=\sqrt{\frac{1}{n-1}\overset{n}{\underset{i=1}{\sum }}{\left(x_{i}-\bar{x}\right)}^{2}}$$
where $x_{i}$ is a value of i-th measurement, n is a number of all measurements and $\bar{x}$ is an arithmetic average of n measurements.

The double-sided confidence interval m for the average value of the population is determined by the double inequality (95% confidence level):(10)$$\bar{x}-\frac{t_{0.975}}{\sqrt{n}}s<m<\bar{x}-\frac{t_{0.975}}{\sqrt{n}}s$$
where $\bar{x}$ is the arithmetic average of n measurements, s is a standard deviation, n is a number of all measurements and $t_{0.975}$ is a quantile of t-Student distribution for the given volume of measurements.

In addition, to assess the mechanism of damage the structure of the broken ends of the cords were observed using the metallurgy microscope and the camera. The results were described only in a qualitative way because the tested cord composition (e.g., chemical) was strictly protected by the manufacturer. 

On the basis of the achieved photos (Figure 11), it can be seen that each cord was built of two different types of fibers, what decided the form of the damaged ends of cords. For cord I (Figure 11a), the change in the diameter can be observed. The damage of the thin fibers (white) appeared on the sides of the cord; however, those fibers were longer at the broken ends, and they likely impacted the cord strength. For cord II (Figure 11b), both broken ends were symmetrical and as in cord I, likely the thin white fibers broke at the end of the tensile test. The fibers were tangled, and the area of braid damage was long. 

For cord III (Figure 11c), the broken ends were symmetric but they broke in accordance to braid angle regularly. Only the ends of the cord IV were not symmetrical (Figure 11d): one of them had short damage area, and the second one had long tangled thin fibers. Such differences between cords were caused by the material properties of each component and the way of braiding. In all cases, the damage was catastrophic.

## 6. Conclusions

The obtained results were compared with the values of elongation at break and breaking force indicated by the manufacturers. The analysis is presented in Table 4. The obtained minimum breaking force was used to compare the achieved results with the manufacturers’ data.

There is a clearly visible difference when measuring the deformation with the extensometer and the results read from the measuring head of the testing machine. This effect, as already indicated, most likely results from the method of fastening the cords (wrapping on the rollers of the grip). The read displacement from the head also considers the behavior of the cord in the rollers, and thus it cannot be used to assess the deformation in the measuring part of the samples. The correct results were obtained using the videoextensometer. The calculated relative error was from 0.7% (Cord II) to 21.5% (Cord I). In addition, despite the strain measurement method, the force was measured correctly. The relative error of measurement in this case was from 0.4% (Cord IV) to 6.2% (Cord II). It should be noted that manufacturers’ data contain the minimum force declared by the producer.

On the basis of the obtained test results, we found that the stand for testing the cords, as well as the adopted research methodology, were correct. This stand can be used to test various types of threads, cords and fibers. The obtained results will be used to build models and simulation calculations of the behavior of single cords and composite layers with rubber cords in various applications (from simple covers to modeling tires for military vehicles and aircraft).

The advantages of the proposed method are as follows: eliminating the influence of the mounting method on the obtained results, increasing the availability of experiments in which the test is conducted with the use of universal tools inserted in place of special tools, and the cord section is released from the additional influences resulting from the contact with the surface of the guiding handle. In the proposed method, a novel testing stage for synthetic cords tensile testing was designed. The method of mounting the cord using rollers allowed the uniform distribution of stress during test neglecting the influence of the machine clamps and gaining the proper form of damage of the tested material.

## Figures and Tables

**Figure 1 materials-15-04163-f001:**
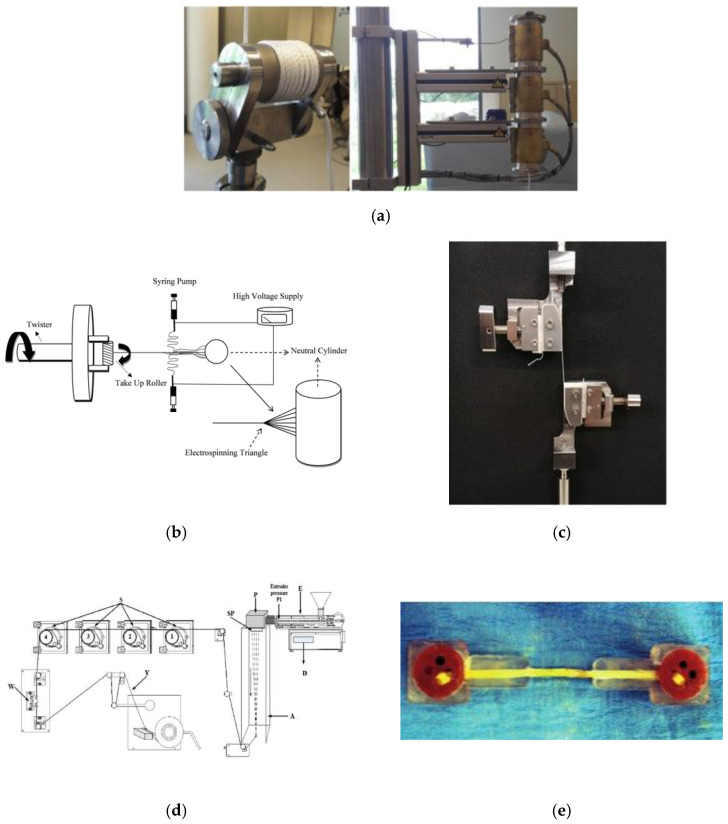
Experimental set-ups: (**a**) for a dynamic test of HMPE at high temperature [23], (**b**) electrospinning tension of polyamide 66 yarns [24], (**c**) gripping systems used for monotonic quasi-static tensile tests [26], (**d**) laboratory spinning equipment COLLIN^®^ CMF 100 [27] and (**e**) cord specimen fixed using epoxy resin molding.

**Figure 2 materials-15-04163-f002:**
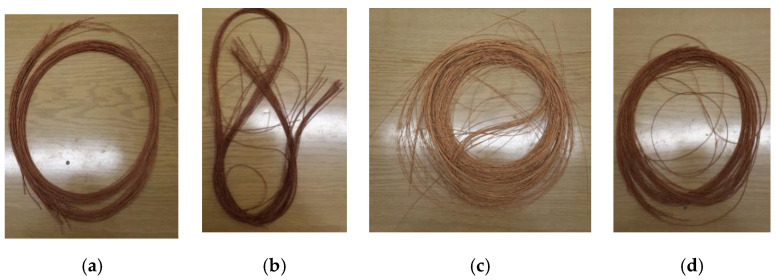
Photos of samples of four materials submitted for testing: (**a**) I—Kordarna Plus PA6; (**b**) II—PA6_Cordenka; (**c**) III—PES 1440; and (**d**) IV—Kordarna Plus PES.

**Figure 3 materials-15-04163-f003:**
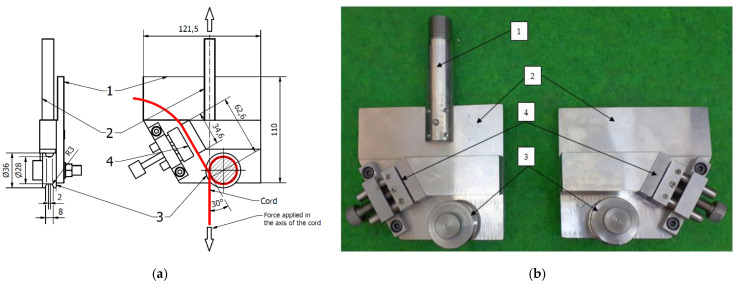
Cord grips: (**a**) technical drawing with cord and applied force visualization and (**b**) real photo; 1—machine jaws with flat inserts; 2—threaded mount; 3—rollers; and 4—clamps of cord grips.

**Figure 4 materials-15-04163-f004:**
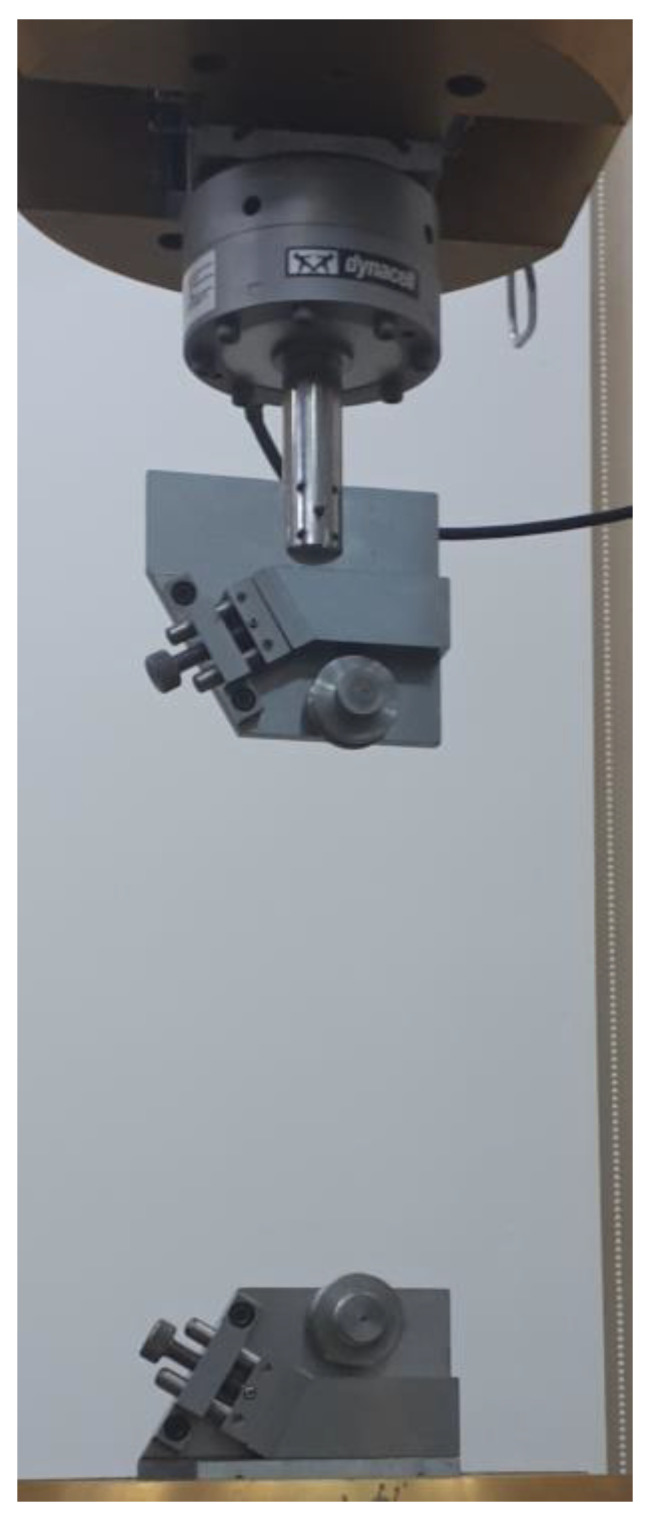
Thread grips installed in the hydraulic clamps of the testing machine with an additionally installed force precise measuring head.

**Figure 5 materials-15-04163-f005:**
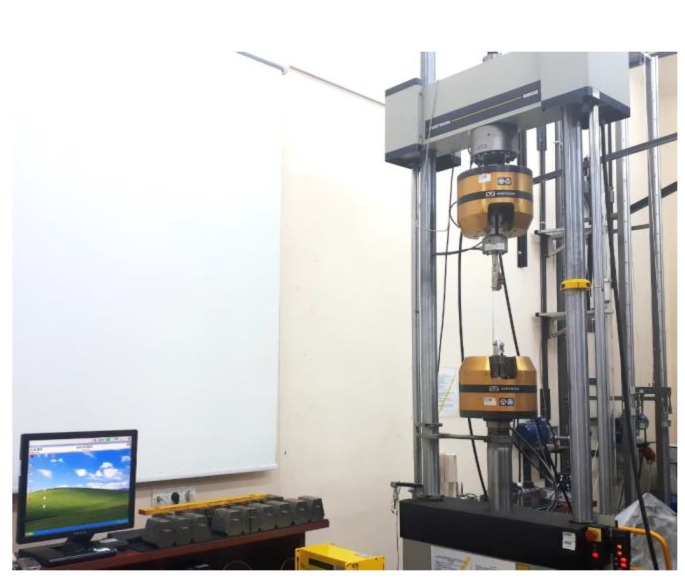
Instron 8802 testing machine for static tests.

**Figure 6 materials-15-04163-f006:**
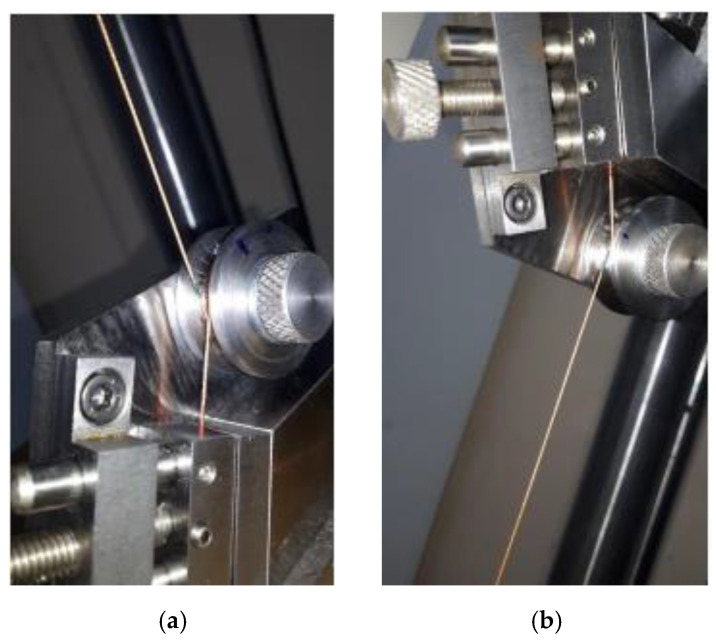
Markings of cord gauge length in the (**a**) lower and (**b**) upper holder.

**Figure 7 materials-15-04163-f007:**
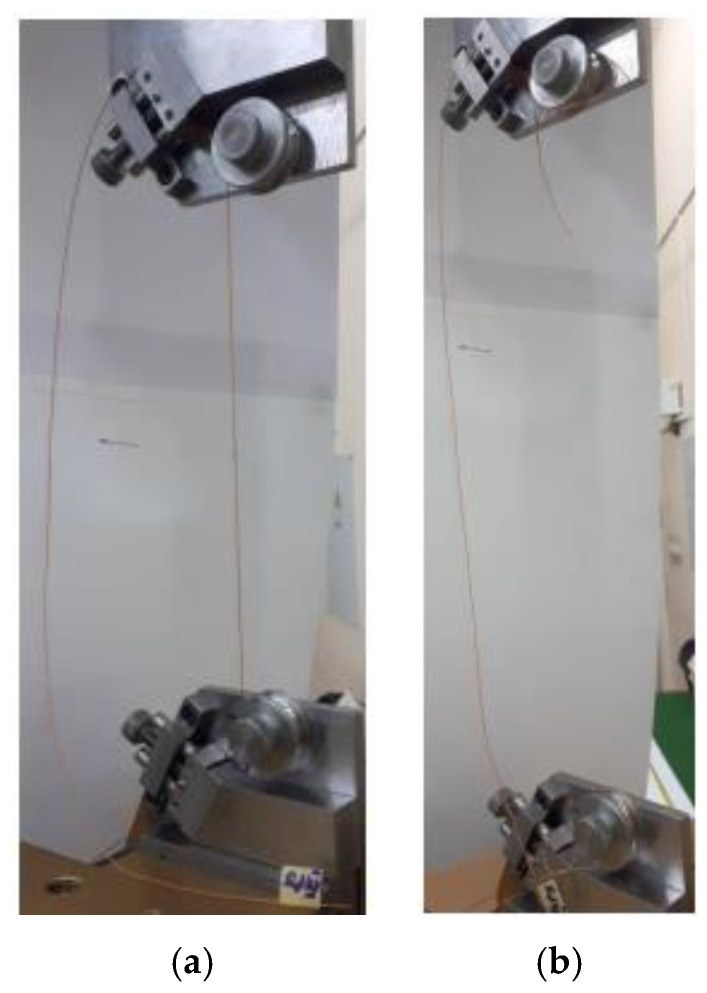
Photos of tested sample of cord III: (**a**) before test and (**b**) after test.

**Figure 8 materials-15-04163-f008:**
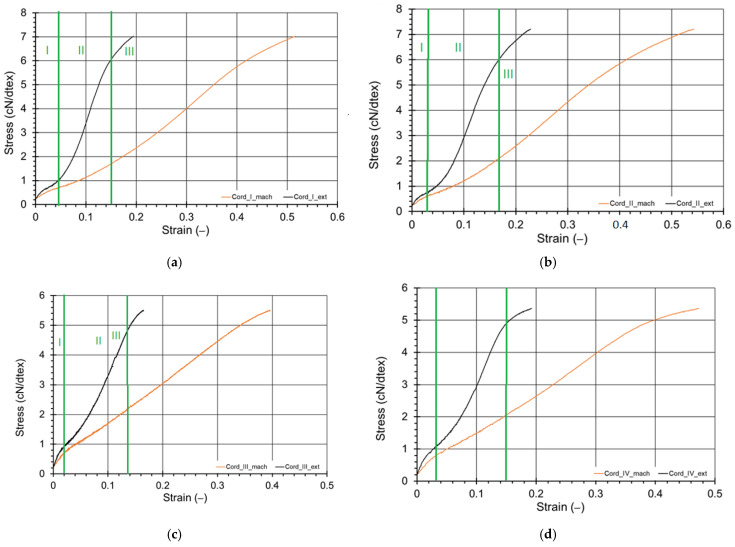
Stress−strain curves for (**a**) cord I, (**b**) cord II, (**c**) cord III and (**d**) cord IV, where “_mach” means the results from the testing machine measurement head, “_ext” −measurement by videoextensometer.

**Figure 9 materials-15-04163-f009:**
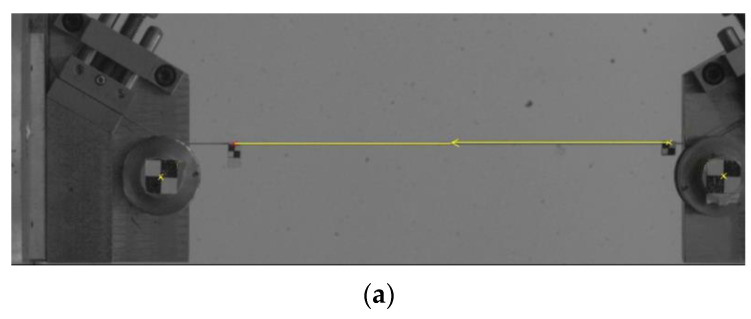
Photos of tested sample of cord I during strain measurement with DIC method: (**a**) at beginning of test, (**b**) halfway through test and (**c**) just before breaking.

**Figure 10 materials-15-04163-f010:**
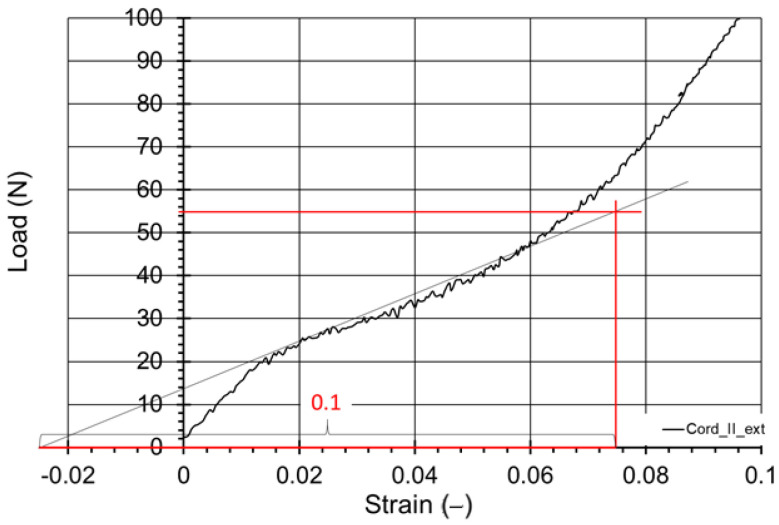
Example of initial modulus calculation for Cord II tested using videoextensometer.

**Figure 11 materials-15-04163-f011:**
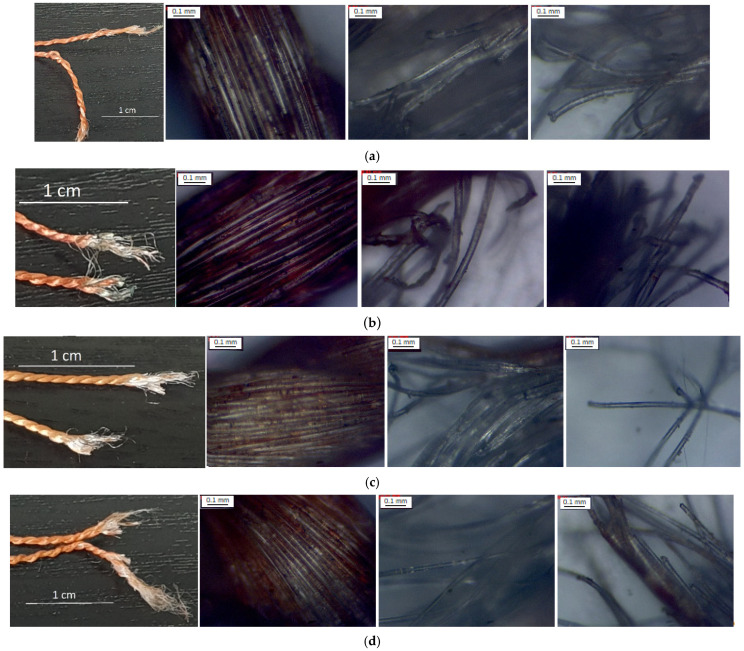
Broken ends of (**a**) cord I, (**b**) cord II, (**c**) cord III and (**d**) cord IV; from left: real photo, microscopic photo of undamaged structure and inside damage area. Broken ends of single threads.

**Table 1 materials-15-04163-t001:** Names and symbols of cord materials used in the tests (manufacturers’ data).

Name	Symbol	Dtex	Doubling	Breaking Force Min. (N)	Elongation for 45N	Elongation at Break	Thickness (mm)	Twist	Initial Force (N)
Kordarna Plus PA6	I	1880	×1×2	265	6.5%	24 ± 2%	0.76	S 335/335	9.40
PA6_Cordenka	II	1880	×1×2	265	6.5%	23 ± 2%	0.76	S 335/335	9.40
PES 1440 (special version)	III	1440	×1×2	173	NA	16 ± 2%	0.65	S 410/410	7.20
Kordarna Plus PES	IV	1670	×1×2	205	3.5%	16 ± 2%	0.70	S 390/390	8.35

The samples were conditioned under the test conditions for more than 10 h. Before the test, they were unpacked and kept under the measurement conditions for at least a few minutes.

**Table 2 materials-15-04163-t002:** Parameters of Instron 8802 testing machine.

Sample Symbol	Data Sampling Rate Instron 8802	Load Speed	Initial Force
	(Hz)	(mm/min)	(N)
All samples	50	300	1.7–2.1

**Table 3 materials-15-04163-t003:** Standard deviation and uncertainty with 95% of confidence of t achieved results.

Standard Deviation	Cord I	Cord II	Cord III	Cord IV
Breaking force (N)	6.8	6.2	3.3	3.1
Elongation at break * (−)	0.0268	0.0162	0.0167	0.0235
Elongation at break ext ** (−)	0.0060			
**Uncertainty (95% of Confidence)**	**Cord I**	**Cord II**	**Cord III**	**Cord IV**
Breaking force (N)	±3.8	±4.4	±2.0	±1.9
Elongation at break * (−)	±0.0148	±0.0116	±0.0101	±0.0142
Elongation at break ext ** (−)	±0.0149			

* based on the displacement measurements using the machine head. ** based on the displacement measurements using the DIC method.

**Table 4 materials-15-04163-t004:** Result analysis.

Name	Symbol	Actual Linear Density (Dtex)	Data Source	Initial Modulus (cN/dtex)	Breaking Force (N) Min./Aver.	Relative Error (%)	Elongation at Break (−)	Relative Error (%)	Strain Correction Factor
									Extensometer/Machine Head
Kordarna Plus PA6	I	4068	Producer’s	-	265/-	-	0.24	-	-
			Machine head	-	274.2/286.3	3.5/8.0	0.50	109.7	-
			Extensometer	13.80	274.2/286.3	3.5/8.0	0.188	−21.5	0.374
PA6_Cordenka	II	4139	Producer’s	-	265/-	-	0.23	-	-
			Machine head	-	281.5/290.5	6.2/9.6	0.54	132.8	-
			Extensometer	13.30	281.5/290.5	6.2/9.6	0.228	−0.70	0.427
PES 1440	III	3382	Producer’s	-	173/-	-	0.16	-	-
			Machine head	-	178.5/184.4	3.2/6.6	0.390	143.7	-
			Extensometer	21.30	178.5/184.4	3.2/6.6	0.165	2.94	0.422
Kordarna Plus PES	IV	3963	Producer’s	-	205/-	-	0.16	-	-
			Machine head	-	205.8/211.8	0.4/3.3	0.456	185.0	-
			Extensometer	19.20	205.8/211.8	0.4/3.3	0.192	20.25	0.422

## Data Availability

Not applicable.

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
