# Peer review of "Experimental Investigation of the Tensile Behavior of Selected Tire Cords Using Novel Testing Equipment"

_materials, 2022, doi:10.3390/ma15124163_

Round 1

Reviewer 1 Report

The manuscript is well written and easy to follow. The description of the proposed device for measuring the mechanical properties is very interesting for overcoming the challenges of performing tensile tests in wires. The main general problem is the poor description in the discussion and conclusion sections of the work. Those need some improvement to be able to be publishable, otherwise, the presented work becomes merely a technical report. I suggest a major revision for the present work in some points below.

1.      Mistyping of “f” in line 66

2.      Fig. 4 is only an image of the rig. To commit to the reproducibility of the work, I suggest drawing a schematic image showing the dimensions of the device as well.

3.      Fig. 8 shows only a broken wire, which does not bring any information for the readers. This is somewhat useless. Instead, you should show microscopy images of the fracture surfaces for the different materials and discuss the fracture mechanism. What is relevant for the scientific community is why those wires behave differently based on your experiments. You must explain it.

4.      Fig. 9 shows force-strain curves. We as readers are interested in stress-strain curves so that one can compare the materials (mechanical properties) with others in the literature. Please, change the y-axis to engineering stress. Also, discuss all stages in the curves.

5.      Also, how about presenting material properties e.g. Young’s modulus, and strengths? To make your new setup reliable you need to compare material properties with other publications or datasheets of the material supplier. Otherwise, presenting only your findings is not enough for a scientific paper. Please, improve it.

6.      From line 203, you are discussing the optical extensometer setup. Shouldn´t it be in the methodology? You are describing it in results and discussion.

7.      How about comparing the force-displacement curves from the grips and from the DIC targets, so that people can really see the difference or improvements in your device. Please discuss more on that. You should argue why your device is good and reliable with more description, writing and comparison, and not only with photos.

Author Response

Thank You for the comments. Please find our answers in the attached file.

Reviewer 2 Report

 The authors describe in this paper in detail and with interesting scientific details the experiment performed with aramid and polyamide cords used in the wide range of applications. Pawel Bogusz et al proposed a novel testing stage mounted in the testing machine clamps to achieve the uniform tensile stress distribution in the gage length of the measured cords. The results of deformations were measured in two ways: using testing machine head displacement and videoextensometer. Stress curves of four distinguish cords were evaluated and compared. The second method allowed to acquire results differing from the manufacturers' data from 0.87 % to 24.17%, which allowed for the conclusion that the designed test stand allows for obtaining reliable results for stretched cords. The problem of the reliable experimental study of tensile behavior of synthetic cords was considered in the paper.

The paper is well structured and the methods are described in a scientific way. Therefore, the paper can be published after a minor revision:

 a) In the INTRODUCTION section, the literature review is not sufficient for a scientific article. You must review more articles in the same field, especially more articles from the MDPI journal.

 b) The novelty of the paper is not clear. Please highlight your novelty.

  c) I do not see the mathematical apparatus in this article, with equations to support the calculations and the experiment performed. Also all your equations must be cited.

  d)How much is your uncertainty error? You must calculate the uncertainty analysis.

 Best regards, 

Author Response

Dear Reviewer,

Please find our answers in the attached file.

Reviewer 3 Report

The authors made Investigation on Tensile Behavior of Selected aramid Yarns Using Novel Testing Equipment. The research sound interesting, however the manuscript need some arrangements in order to be accepted for publication.

In the abstract the authors should be more specific about the reasons why they are doing this study. There is something wrong in the test method for testing aramid yarns

In line 46 the authors should review the sentence; I believe that is missing figure 1. The authors have to review the reference also.

The authors have to avoid sentences as:

preparation on their mechanical properties was shown in [11].

 The PVA (poly(vinyl acetate)) braided yarns mechanical behavior using gripping systems (Fig. 2c) was tested in [19]”

“using laboratory spinning equipment COLLIN® CMF 100 – in [20]”

The authors should put the group who developed the investigation and after the references

Please review the sentence in line 66 “testing f yarns”

The problems to the authors referring in paragraph (line 78-88) could be solved following the guidance of ASTM D4018.

The figures should be cited in the manuscript progressively (1, 2, 3, 4..). The authors have cited the figure 4 before the figure3. The authors have to fix this mistake

Please fix the figure 4 the numbers and arrows are moved

Please review sentence of line 144 “on the rollers of the rollers”

 The authors should provide the linear density of each specimen tested

The authors should explain in detail how was performed the DIC method and why differ so much from universal testing Machine measurements.

The authors do not give convincing arguments of different results obtained by both techniques

I recommend to measure the displacement of the machine with a dial indicator (extensometer) and if the machine give values similar to extensometer then the authors have to review the DIC method. Because sometimes the providers give wrong values of their products.

Another suggestion is:

To avoid this situation “The read displacement from the head also takes into account the behavior of the cord in the rolls, so it cannot be used to assess the deformation in the measuring part of the samples.” The authors should test the method recommended in ASTM D4018

Author Response

(The authors gave the same response as above.)

Round 2

Reviewer 1 Report

·        Just for my own sake, why Eqs. 1 – 4 for a 3D formulation is being used since your measurements are basically in 2D (or maybe 1D, distance between the targets). Please, do not use unnecessary things. The equations should correspond the problem you are addressing in this work. I think it would also be wise to label the points A and B in Fig. 5. This brings originality!

·        Please, the units of stress are given in Pa (or typically MPa). Fix that in Fig. 9.

·        The unity of the elastic modulus is typically given in GPa. Please fix that in Table 4.

Author Response

Thank You for the comment. We understand Your point of view, and we also deeply studied the subject of units for cord testing. The problem in showing the data in Pa is the reliable measurement of  diameter of cord, which is built of braided thin yarns. So the stress for cords is often presented in the literature in N/tex to avoid the problems shown above (we citied one of such papers in the text). Also tex is the common unit for cord linear density in textile industry.

Thank you once again, and we hope that our explanation will meet your doubts.

Reviewer 3 Report

no one

Author Response

Thank You for the acceptance of our work!